# PDF-1 neuropeptide signaling regulates sexually dimorphic gene expression in shared sensory neurons of *C. elegans*

**Zoë A Hilbert, Dennis H Kim***

Department of Biology, Massachusetts Institute of Technology, Cambridge, United States

**Abstract** Sexually dimorphic behaviors are a feature common to species across the animal kingdom, however how such behaviors are generated from mostly sex-shared nervous systems is not well understood. Building on our previous work which described the sexually dimorphic expression of a neuroendocrine ligand, DAF-7, and its role in behavioral decision-making in *C. elegans* (Hilbert and Kim, 2017), we show here that sex-specific expression of *daf-7* is regulated by another neuroendocrine ligand, Pigment Dispersing Factor (PDF-1), which has previously been implicated in regulating male-specific behavior (Barrios et al., 2012). Our analysis revealed that PDF-1 signaling acts sex- and cell-specifically in the ASJ neurons to regulate the expression of *daf-7*, and we show that differences in PDFR-1 receptor activity account for the sex-specific effects of this pathway. Our data suggest that modulation of the sex-shared nervous system by a cascade of neuroendocrine signals can shape sexually dimorphic behaviors.
DOI: https://doi.org/10.7554/eLife.36547.001

## Introduction

Behavioral differences between the sexes of animal species can make major contributions to the reproductive fitness of the organism. While sex-specific behaviors can be readily observed, the mechanistic basis of such behavioral differences is less well understood. Morphological differences, including the existence of sex-specific neurons, have been documented in the nervous systems of many species, but differences in sex-shared neurons have also been implicated in generating sex-specific behaviors. In particular, how sex-specific behavioral circuits are generated within the features of the nervous system common to both sexes has been the focus of recent studies in diverse organisms. Studies of the mouse vomeronasal organ (VNO) has suggested that the functional circuits for both male- and female-specific behaviors such as courtship and aggression are intact in the brains of both sexes and are modulated by VNO activity in response to pheromone cues (*Kimchi et al., 2007*; *Stowers et al., 2002*). In a similar vein, the *Drosophila* male pheromone 11-*cis* Vaccenyl acetate (cVa) has been shown to be sensed by the same neurons in the two sexes but stimulates distinct sex-specific behavioral responses (*Datta et al., 2008*; *Kohl et al., 2013*; *Kurtovic et al., 2007*; *Ruta et al., 2010*). These examples and others have provided some insight into the sexual dimorphisms present in the nervous system and their contributions to behavior, although many open questions remain (*Dulac and Kimchi, 2007*; *Stowers and Logan, 2010*; *Yang and Shah, 2014*).

In the nematode *Caenorhabditis elegans*, behavioral differences between the two sexes—hermaphrodites and males—range from behaviors exclusively performed by one sex, such as egg laying by hermaphrodites and the mating program of males (*Liu and Sternberg, 1995*), to those in which the two sexes differ in their responses to the same stimuli, including differing responses to pheromone (*Fagan et al., 2018*; *Jang et al., 2012*; *Srinivasan et al., 2008*), food-related cues (*Ryan et al., 2014*), and conditioning to aversive stimuli (*Sakai et al., 2013*; *Sammut et al., 2015*).

***For correspondence:**
dhkim@mit.edu

**Competing interests:** The authors declare that no competing interests exist.

While sex-specific neurons regulate corresponding behaviors in *C. elegans*, the 294 neurons that are common to the nervous systems of both hermaphrodites and males have emerged as major contributors to a number of different sexually dimorphic behaviors (*Barr et al., 2018*; *Barrios et al., 2012*, *2008*; *Fagan et al., 2018*; *Lee and Portman, 2007*; *Mowrey et al., 2014*; *Sakai et al., 2013*). In particular, recent work has uncovered sexually dimorphic differences in axonic and dendritic morphology and synaptic connectivity within the sex shared nervous system, which can modulate neuronal circuits and behavior (*Hart and Hobert, 2018*; *Oren-Suissa et al., 2016*; *Serrano-Saiz et al., 2017a*; *Weinberg et al., 2018*). In addition, studies of sexually dimorphic gene expression (*Hilbert and Kim, 2017*; *Ryan et al., 2014*; *Serrano-Saiz et al., 2017a*) and neurotransmitter identity (*Gendrel et al., 2016*; *Pereira et al., 2015*; *Serrano-Saiz et al., 2017a*, *2017b*) have suggested that sexual differentiation of neurons within the sex-shared nervous system of *C. elegans* is critical for the establishment of sexually dimorphic behaviors.

We have previously demonstrated that *daf-7*, which encodes a TGFβ family neuroendocrine ligand that regulates diverse aspects of *C. elegans* behavior and physiology (*Chang et al., 2006*; *Fletcher and Kim, 2017*; *Gallagher et al., 2013*; *Greer et al., 2008*; *Milward et al., 2011*; *Ren et al., 1996*; *Shaw et al., 2007*; *White and Jorgensen, 2012*; *You et al., 2008*), is expressed in a sex-specific and context-dependent manner in the sex-shared ASJ chemosensory neurons and functions to promote exploratory behaviors (*Hilbert and Kim, 2017*; *Meisel et al., 2014*). Regulation of *daf-7* expression in the ASJ neurons requires the integration of sensory and internal state information including the sex and age of the animal, its nutritional state, and the type of bacterial species it encounters in its environment (*Hilbert and Kim, 2017*). These stimuli feed into the regulation of *daf-7* expression in the two ASJ neurons in a hierarchical manner, which enables the animal to make behavioral decisions taking into account past experiences as well as its current environment.

Here, we report the identification of a second neuroendocrine signaling pathway, the Pigment Dispersing Factor (PDF-1) pathway, which functions to regulate the expression of *daf-7* and its effects on behavior in a sex-specific manner. We show that PDF-1 pathway signaling, which has previously been shown to be essential for male mate-searching behavior (*Barrios et al., 2012*), functions sex-specifically in the ASJ neurons themselves to regulate *daf-7* expression. Further, we demonstrate that the sex-specificity of PDF-1 regulation of *daf-7* derives from differences in the activation of PDF-1 signaling downstream of the PDF-1 receptor gene, *pdfr-1*, in the ASJ neurons. Our data suggest that the gating of neuronal responses to neuropeptide modulators through sex-specific restriction of receptor activity is a mechanism by which sex-specific behaviors can be generated from the largely sex-shared nervous system of *C. elegans*.

## Results and discussion

### PDF-1 neuropeptide signaling regulates the sex-specific expression of *daf-7* in the ASJ chemosensory neurons

To explore the molecular and genetic mechanisms that underlie the sex-specificity of *daf-7* expression, we identified a number of candidate genes that had previously been shown to be involved in the regulation of mate-searching behavior or other aspects of male physiology and tested mutants of these genes for effects on *daf-7* expression in the male ASJ neurons. Through this approach, we identified the PDF-1 neuropeptide signaling pathway as a regulator of *daf-7* expression in the ASJ neurons (*Figure 1*). The PDF neuropeptide signaling pathway is conserved among insects, crustaceans and nematodes. In *Drosophila melanogaster*, PDF signaling has been well studied for its critical role in the regulation of circadian rhythmicity (*Helfrich-Förster, 1995*; *Park and Hall, 1998*; *Park et al., 2000*; *Renn et al., 1999*), but it has also been shown to modulate geotaxis (*Mertens et al., 2005*), pheromone production and mating behaviors (*Fujii and Amrein, 2010*; *Kim et al., 2013*; *Krupp et al., 2013*). In *C. elegans*, PDF-1 signaling has been established as an important regulator of locomotion, roaming behaviors, quiescence, and notably, male mate-searching behavior (*Figure 1A*; *Barrios et al., 2012*; *Choi et al., 2013*; *Flavell et al., 2013*; *Janssen et al., 2008*, *2009*; *Meelkop et al., 2012*).

We observed that males with mutation of either the PDF-1 neuropeptide ligand or its receptor, PDFR-1, have markedly attenuated expression of *daf-7* in the ASJ neuron pair (*Figure 1B*). The ASI

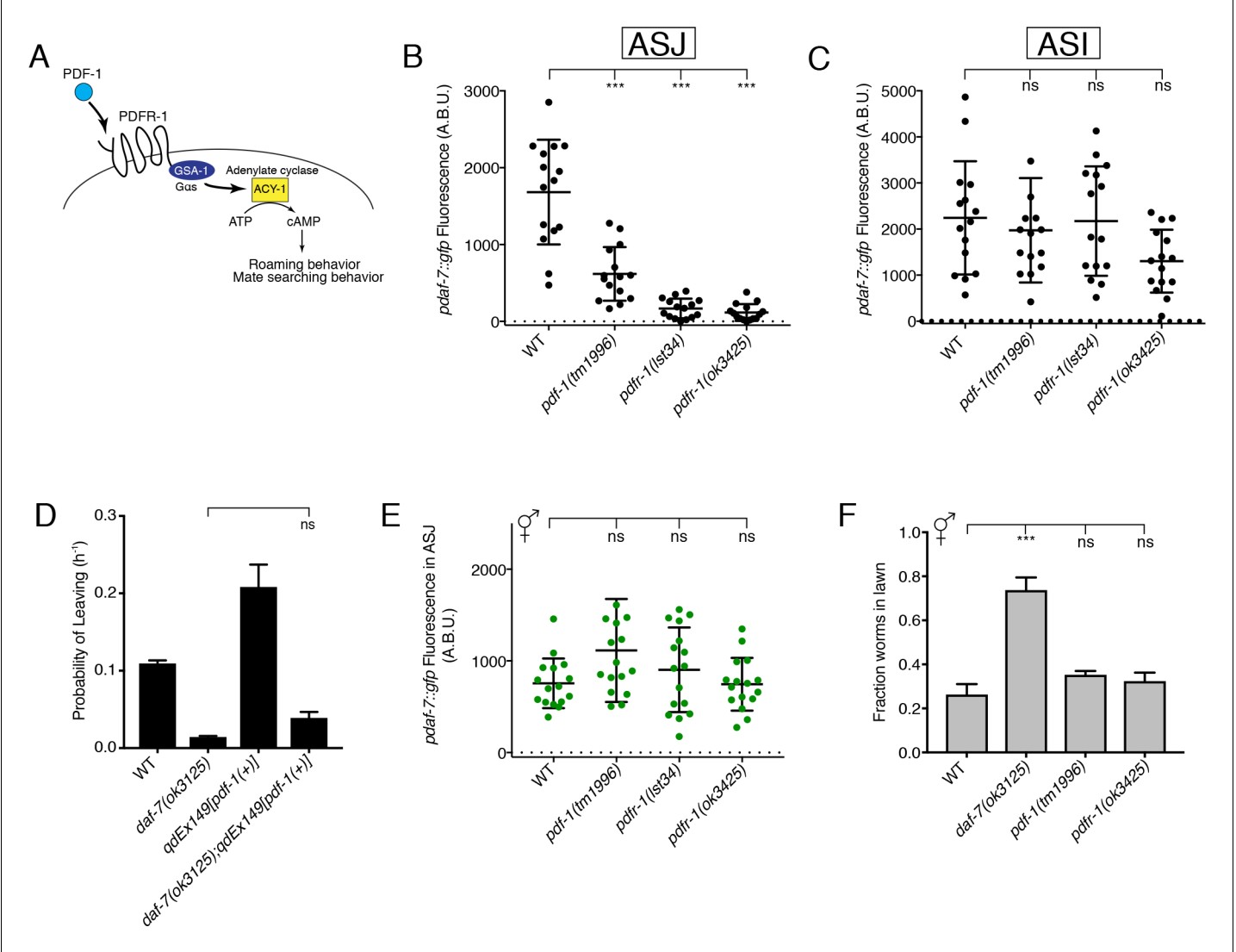

**Figure 1.** The PDF-1 pathway is required for the male-specific expression of *daf-7* in the ASJ neurons and its effects on male mate-searching behavior. (A) PDF-1 signaling activates cAMP production and regulates both roaming behavior and male mate-searching behavior in *C. elegans*. (B–C) Maximum fluorescence values of *pdaf-7::gfp* in the ASJ (B) and ASI (C) neurons of adult male animals. ***$p < 0.001$ as determined by ordinary one-way ANOVA followed by Dunnett's multiple comparisons test. Error bars represent standard deviation (SD). ns, not significant. n = 15 animals for all genotypes. (D) Probability of leaving values for epistasis experiment between *daf-7(ok3125)* and a PDF-1 overexpressing line. Values plotted are the mean +SEM for three independent experiments. Significance determined by unpaired t-test with Welch's correction. ns, not significant. n = 60 total animals for all genotypes except the *daf-7(ok3125); qdEx149* strain where n = 48. (E) Maximum fluorescence values of *pdaf-7::gfp* in the ASJ neurons of hermaphrodites after 16 hr on *P. aeruginosa*. Significance determined by ordinary one-way ANOVA followed by Dunnett's multiple comparisons test. Error bars represent SD. ns, not significant. n = 15 animals for all genotypes. (F) Lawn occupancy of animals on *P. aeruginosa* after 16 hr. ***$p < 0.001$ as determined by ordinary one-way ANOVA followed by Dunnett's multiple comparisons test. Values plotted indicate the mean + SD for three replicates. Number of animals assayed are as follows: WT (n = 89), *daf-7* (n = 66), *pdf-1* (n = 117), *pdfr-1* (n = 105).

DOI: https://doi.org/10.7554/eLife.36547.002

The following figure supplements are available for figure 1:

**Figure supplement 1.** The PDF-1 pathway regulates mate-searching behavior via male-specific modulation of *daf-7* expression in ASJ and through parallel mechanisms.

DOI: https://doi.org/10.7554/eLife.36547.003

**Figure supplement 2.** PDF-1 pathway mutant males can upregulate *daf-7* expression in ASJ in response to *P. aeruginosa* exposure and show intact pathogen avoidance behavior.

DOI: https://doi.org/10.7554/eLife.36547.004

chemosensory neurons are established sites of *daf-7* expression in both male and hermaphrodite animals (*Ren et al., 1996*; *Schackwitz et al., 1996*), so we asked if the PDF-1 signaling pathway also regulates *daf-7* expression in these neurons. In the PDF-1 pathway mutant males, we observe no difference in *daf-7* levels in the ASI neurons when compared to WT (*Figure 1C*), suggesting that the PDF-1 pathway specifically affects the regulation of *daf-7* in the ASJ neuron pair.

Expression of *daf-7* in the ASJ neuron pair of males is required for the male-specific mate-searching behavioral program (*Hilbert and Kim, 2017*), while the PDF-1 pathway has similarly been implicated as a regulator of this same behavior (*Barrios et al., 2012*). Given the role that this PDF-1 pathway plays in regulating the expression of *daf-7*, we set out to determine if the effects of the PDF-1 pathway on mate-searching behavior are the result of PDF-1 and DAF-7 functioning through a single pathway or through separate parallel pathways. Overexpression of the *pdf-1* genomic sequence confers increased mate-searching behavior in male animals (*Figure 1D*; *Barrios et al., 2012*). We introduced a *daf-7* mutation into these transgenic PDF-1 overexpressing lines and observed that the effect of PDF-1 overexpression on mate-searching behavior was suppressed by loss of *daf-7* function (*Figure 1D* and *Figure 1—figure supplement 1A*). However, we observed that overexpression of *daf-7* in the ASJ neurons of *pdf-1(tm1996)* mutant males could not rescue the mate-searching defects of these animals (*Figure 1—figure supplement 1B*). Taken together, the results of this epistasis analysis suggest that PDF-1 regulates mate-searching behavior in males through the regulation of *daf-7* expression in the ASJ neurons and through additional mechanisms functioning in parallel to DAF-7 signaling.

We have previously reported that *daf-7* expression serves a dual role in the ASJ neurons, functioning in males to promote food-leaving behaviors (*Hilbert and Kim, 2017*), but also being induced by the presence of *Pseudomonas aeruginosa* in both sexes to promote pathogen avoidance behaviors (*Meisel et al., 2014*). Given this and our interest in identifying male-specific regulators of *daf-7* expression, we asked if the PDF-1 pathway is required for the upregulation of *daf-7* expression in response to *P. aeruginosa*. We did not observe a requirement for PDF-1 signaling in the induction of *daf-7* expression in the ASJ neurons after 16 hr on *P. aeruginosa*; both *pdf-1* and *pdfr-1* mutant hermaphrodites had equivalent levels of *daf-7* expression when compared to control animals (*Figure 1E*). Similarly, males that are mutant for either the PDF-1 ligand or receptor (and show little to no *daf-7* expression in their ASJ neurons on *E. coli*, see *Figure 1B*) were capable of upregulating *daf-7* expression in ASJ upon exposure to *P. aeruginosa* (*Figure 1—figure supplement 2A*). Given the previously established function of *daf-7* expression in the ASJ neurons of hermaphrodites in promoting pathogen avoidance behavior (*Meisel et al., 2014*), these results predict that mutants in the PDF-1 pathway should have no defects in their ability to avoid a lawn of pathogenic *P. aeruginosa*. Consistent with this expectation, we observed that while *daf-7* mutant hermaphrodites fail to avoid a lawn of pathogenic bacteria, the *pdf-1* and *pdfr-1* mutant hermaphrodites appear wild-type for their ability to perform this behavior (*Figure 1F*). Similarly, *pdf-1* and *pdfr-1* mutant males also displayed robust pathogen avoidance despite their defects in the male-specific mate searching behavior (*Figure 1—figure supplement 2B*; *Barrios et al., 2012*). These data suggest that the PDF-1 signaling pathway acts sex-specifically to regulate *daf-7* expression in the ASJ neurons and its effects on downstream sexually dimorphic behavioral programs.

## PDF-1 signaling acts cell-autonomously in the ASJ neurons to promote *daf-7* expression

The PDF-1 neuropeptide ligand is secreted from multiple neurons in the head region of the animal where a similarly large number of neurons express the PDFR-1 receptor (*Barrios et al., 2012*; *Janssen et al., 2009*; *Meelkop et al., 2012*). To identify the relevant site of action for this pathway in the regulation of *daf-7* expression in males, we used the *pdfr-1(ok3425)* mutant animals and introduced *pdfr-1* cDNA transgenes into specific neurons using heterologous cell-specific promoters. We observed that while the *pdfr-1* mutant males lack *daf-7* expression in the ASJ neurons, introduction of a genomic DNA fragment carrying the *pdfr-1* locus fully rescued this phenotype and restored *daf-7* expression in the ASJ neurons (*Figure 2A and B*). Furthermore, we observed that expression of *pdfr-1* under the control of the ASJ-specific *trx-1* promoter was sufficient to rescue *daf-7* expression in the ASJ neurons of the mutant male animals, suggesting that PDF-1 signals to its receptor, PDFR-1, in the ASJ neurons to influence *daf-7* expression specifically in the male (*Figure 2A and B*).

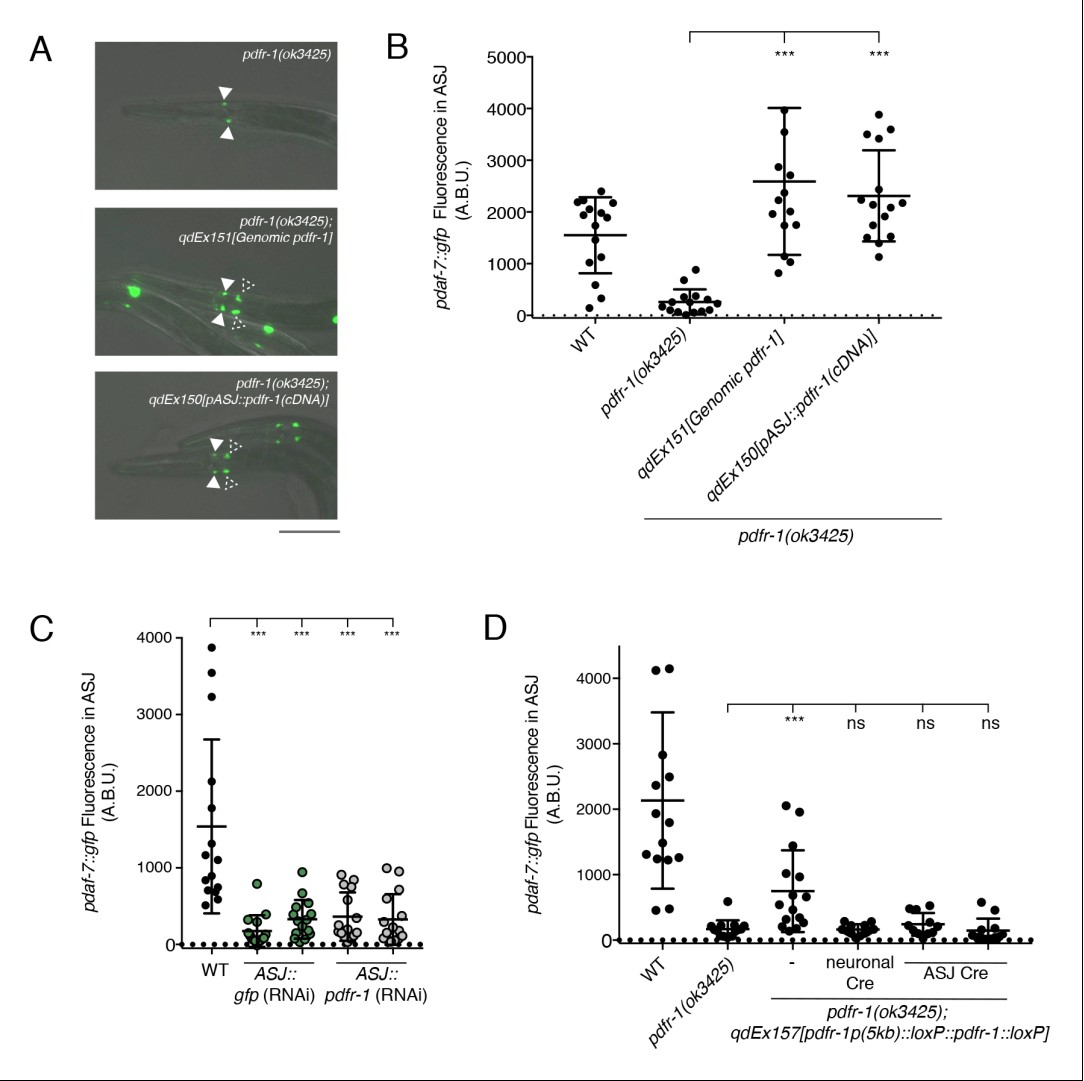

**Figure 2.** PDF-1 signaling is necessary and sufficient in the ASJ neurons for the regulation of *daf-7* expression in male *C. elegans*. (**A**) *pdaf-7::gfp* expression in *pdfr-1(ok3425)* mutant (top), genomic rescue (middle), and ASJ-specific rescue (bottom) animals. Filled arrowheads indicate the ASI neurons; dashed arrowheads indicate the ASJ neurons. Scale bar indicates 50 μm. (**B**) Maximum fluorescence values of *pdaf-7::gfp* in the ASJ neurons of *pdfr-1* rescue males. ***p<0.001 as determined by ordinary one-way ANOVA followed by Dunnett's multiple comparisons test. Error bars represent SD. n = 15 animals for all genotypes. (**C**) Maximum fluorescence values of *pdaf-7::gfp* in the ASJ neurons of WT males (black) and animals with ASJ-specific RNAi of either GFP (green) or *pdfr-1*(gray). ***p<0.001 as determined by ordinary one-way ANOVA followed by Dunnett's multiple comparisons test. Error bars indicate SD. n = 15 animals for all conditions. (**D**) Maximum fluorescence values of *pdaf-7::gfp* in the ASJ neurons of WT, *pdfr-1(ok3425)* mutants, and animals with floxed *pdfr-1* rescued under the control of a 5 kb distal reporter as reported in *Flavell et al. (2013)* (left three columns). *pdfr-1* function was removed either in all neurons or specifically in ASJ using cell-specific expression of *Cre* recombinase (right three columns). ***p<0.001 as determined by ordinary one-way ANOVA followed by Dunnett's multiple comparisons test. ns, not significant. Error bars indicate SD. n = 12–15 animals for each condition.
DOI: https://doi.org/10.7554/eLife.36547.005

To assess the necessity of *pdfr-1* function in ASJ for the regulation of *daf-7* expression in males, we knocked down *pdfr-1* expression in the ASJ neurons via cell-specific RNAi (*Esposito et al., 2007*). We observed that animals with RNAi targeting *pdfr-1* in the ASJ neurons exhibited reduced *daf-7* expression in ASJ comparable to what we observed with ASJ-specific RNAi of GFP, our positive control (*Figure 2C*). Additionally, we generated transgenic animals carrying a floxed copy of the

*pdfr-1* cDNA under the control of a 5 kb region of the endogenous *pdfr-1* promoter, which has been previously reported to rescue roaming behaviors in the *C. elegans* hermaphrodite (*Flavell et al., 2013*). We observed that this construct partially rescued *daf-7* expression in the ASJ neurons of *pdfr-1(ok3425)* mutant males, and ASJ-specific expression of the Cre recombinase suppressed the rescuing effects of this transgene (*Figure 2D*), strongly suggestive that PDFR-1 activity in the ASJ neurons is required for *daf-7* expression. These results indicate that the PDF-1 signaling pathway functions cell-autonomously in the ASJ neuron pair to regulate the sexually dimorphic expression of *daf-7*.

The PDFR-1 receptor is a secretin-family G-protein coupled receptor (GPCR), which has been shown to stimulate Gαs signaling and upregulation of cAMP production in transfected cells as well as in both *Drosophila melanogaster* and *C. elegans* neurons (*Figure 1A*; *Flavell et al., 2013*; *Hyun et al., 2005*; *Janssen et al., 2008*; *Lear et al., 2005*; *Mertens et al., 2005*; *Shafer et al., 2008*). Using a gain-of-function variant of the adenylate cyclase, ACY-1 (*Flavell et al., 2013*; *Saifee et al., 2011*; *Schade et al., 2005*), we asked if activation of the pathway downstream of PDFR-1 specifically in ASJ was sufficient to rescue the defects in *daf-7* expression that we observe in the *pdfr-1* mutant males. We observed that in *pdfr-1* mutant males with transgenic expression of the *acy-1(gf)* cDNA only in the ASJ neurons, *daf-7* expression was fully rescued (*Figure 3A*). This ability to bypass PDFR-1 by activation of cAMP production specifically in the ASJ neuron pair further suggest that the PDF-1 signaling pathway acts directly on the ASJ neurons in order to regulate *daf-7* expression in male animals. Given the hierarchical nature of the regulation of *daf-7* expression in the male ASJ neurons (*Hilbert and Kim, 2017*), we asked if *acy-1* function may serve a broader role coordinating the many inputs of this hierarchy into changes in *daf-7* expression. We previously reported that starvation of adult male animals efficiently suppresses *daf-7* expression in the ASJ neurons in wild-type animals (*Hilbert and Kim, 2017*; *Figure 3B*). Notably, ASJ-specific expression of the *acy-1(gf)* variant in starved males did not suppress the effects of starvation on *daf-7* expression in ASJ (*Figure 3B*), suggestive that the effects of starvation act downstream of or in parallel to PDFR-1-ACY-1 signaling.

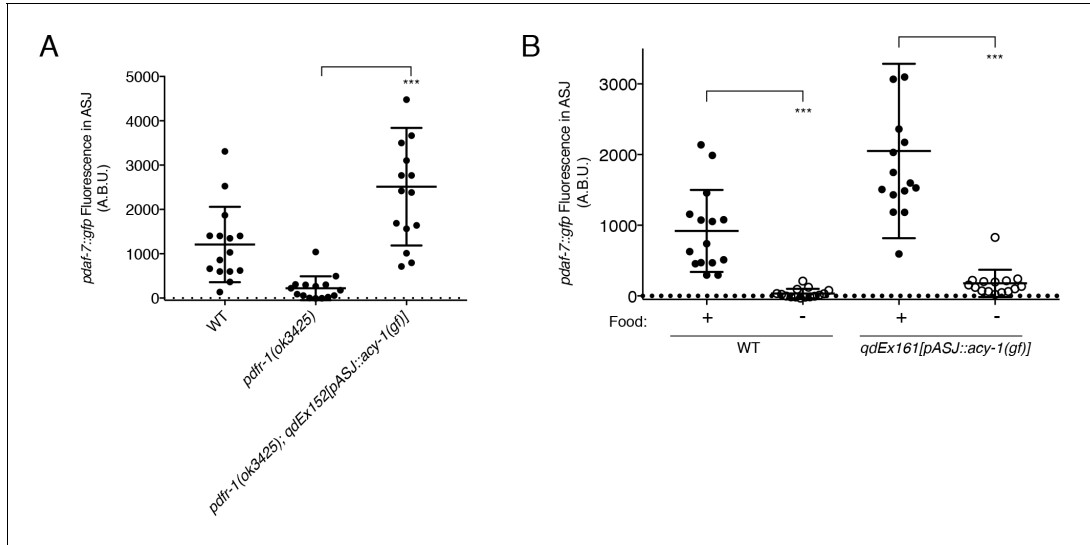

**Figure 3.** ACY-1 acts downstream of PDFR-1 to regulate *daf-7* expression in male ASJ neurons. (**A**) Maximum fluorescence values of *pdaf-7::gfp* in the ASJ neurons of males expressing the gain-of-function ACY-1(P260S) cDNA specifically in ASJ. ***p<0.001 as determined by unpaired t-test with Welch's correction. Error bars represent SD. n = 15 animals for all genotypes. (**B**) Maximum fluorescence values of *pdaf-7::gfp* in the ASJ neurons of WT and ASJ-specific *acy-1(gf)* expressing fed (filled circles) and starved (open circles) males. ***p<0.001 as determined by unpaired t-test with Welch's correction. Error bars indicate SD. n = 15 animals for all conditions.
DOI: https://doi.org/10.7554/eLife.36547.006

## Sex differences in PDF-1 receptor activity underlie the sex-specific regulation of *daf-7* transcription in ASJ

The sex-specificity of the effects of the PDF-1 pathway on *daf-7* regulation and mate searching behavior is intriguing given very little evidence of differences in the expression or function of this neuropeptide pathway between the two *C. elegans* sexes (*Barrios et al., 2012*; *Janssen et al., 2008*, *2009*). It was recently shown that *pdf-1* is produced by the newly identified male-specific MCM neurons and is required for the regulation of sex-specific learning in males, but interestingly, ablation of these neurons has no effect on mate-searching behavior (*Sammut et al., 2015*). Nevertheless, we wondered if there might be unidentified sex differences in the signaling or expression of this PDF-1 neuropeptide pathway in neurons such as ASJ, which would confer its sex-specific effects on the regulation of *daf-7* gene expression. To this end, we asked if activation of the PDF-1 signaling pathway in the ASJ neurons of hermaphrodites might be sufficient to drive *daf-7* expression inappropriately in these animals. We first looked at hermaphrodite animals carrying the same ASJ-expressed *acy-1(gf)* transgene and observed significant upregulation of *daf-7* expression in the ASJ neurons of these hermaphrodites (*Figure 4A*). We next asked whether we could observe *daf-7* expression in the ASJ neurons of hermaphrodite animals with heterologous expression of *pdfr-1* in only the ASJ neurons. Strikingly, we found that in hermaphrodites with overexpression of *pdfr-1* cDNA in the ASJ neurons, *daf-7* expression was also upregulated similar to what we observed in the *acy-1(gf)* transgenic strains (*Figure 4B*). We also quantified *daf-7* expression in the ASJ neurons of hermaphrodites carrying the genomic *pdfr-1* fragment with all of the endogenous regulatory sequence and observed no upregulation of expression in those animals. This control suggests that *daf-7* expression in ASJ cannot be triggered simply as the result of overexpression of *pdfr-1* (*Figure 4B*), rather, these results suggest that expression of PDFR-1 specifically in the hermaphrodite ASJ neurons is sufficient to allow *daf-7* expression in these neurons.

We note that establishing the neuronal expression pattern of *pdfr-1* has been challenging because of the apparent complexity of defining putative regulatory regions of the gene (*Barrios et al., 2012*; *Flavell et al., 2013*; *Janssen et al., 2008*). We sought to examine the transcription of *pdfr-1* in male and hermaphrodite animals using fluorescence in situ hybridization (FISH). We generated fluorescent probes for a region of the *pdfr-1* coding sequence that is shared among all isoforms and verified the specificity of these probes for the *pdfr-1* coding sequence by examining expression in the *pdfr-1(ok3425)* deletion mutant, where we observed no fluorescent signal (*Figure 4—figure supplement 1B*), and in our ASJ-specific rescue lines, where we could only observe fluorescence in the ASJ neurons (*Figure 4—figure supplement 1D*). Imaging of *pdfr-1* transcription in WT animals revealed a diffuse expression pattern with fluorescent signal observable in muscle tissue as well as in neurons, but with few cells having strong signal and many cells with only scattered fluorescent spots, including the ASJ neurons (*Figure 4—figure supplement 1A*). To corroborate and confirm these observations, we also imaged *pdfr-1* transcripts in animals carrying our genomic rescuing fragment, which amplified probe fluorescence throughout the nervous system and muscle (*Figure 4—figure supplement 1C*). We expect that because of the intact endogenous regulatory sequence on this genomic fragment, the mRNA localization we observe in this strain should still be representative of the wild-type expression pattern of *pdfr-1*. While we observe qualitative differences in the abundance of *pdfr-1* mRNA in the ASJ neurons between males and hermaphrodites, we did not detect *pdfr-1* mRNA in all male animals examined. We observed puncta of *pdfr-1* mRNA in the ASJ neurons of about 20% of adult male animals (*Figure 4—figure supplement 1E and F*), whereas similarly aged hermaphrodites did not have a corresponding subpopulation of animals with puncta of *pdfr-1* mRNA. Further analysis of the expression pattern of *pdfr-1* will be required to definitively identify any sexual dimorphism in the expression of this gene. Nevertheless, our functional studies demonstrating that PDFR-1 expression in the ASJ neurons is both necessary and sufficient for *daf-7* expression suggest that the expression or activity of the PDFR-1 receptor may be regulated in a sexually dimorphic manner in these neurons.

## The PDF-1-DAF-7 neuroendocrine signaling cascade regulates sex-specific behavior through the sex-shared ASJ neurons

Building on our previous work on the sexually dimorphic regulation of the neuroendocrine gene *daf-7* and its role in promoting male decision-making behaviors (*Hilbert and Kim, 2017*), we have

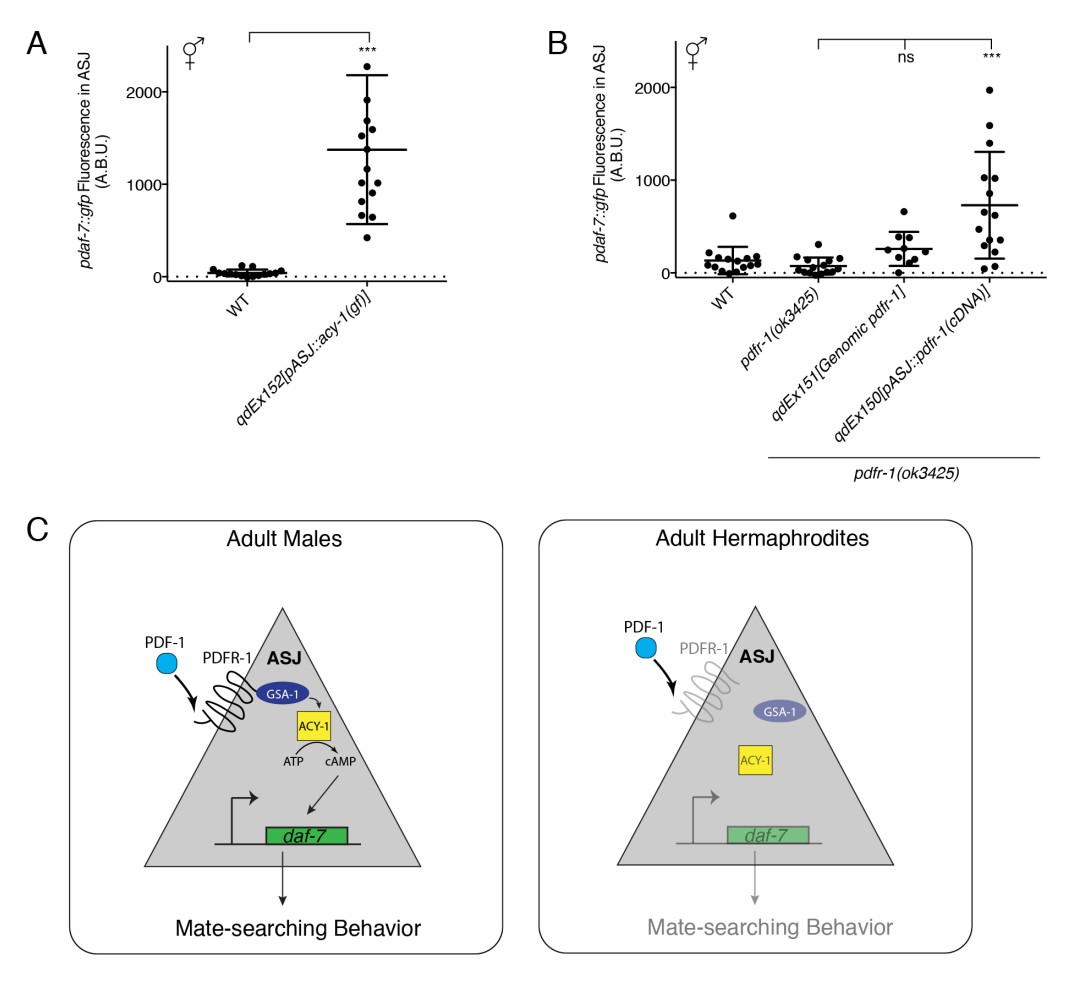

**Figure 4.** Heterologous activation of the PDF-1 pathway in ASJ is sufficient to activate *daf-7* transcription in adult hermaphrodites. (**A**) Maximum fluorescence values of *pdaf-7::gfp* in the ASJ neurons of WT hermaphrodites and hermaphrodites where ACY-1 has been activated (via the gain of function P260S mutant) specifically in ASJ. ***p<0.001 as determined by unpaired t-test with Welch's correction. Error bars represent SD. n = 15 animals for both genotypes. (**B**) Maximum fluorescence values of *pdaf-7::gfp* in the ASJ neurons of hermaphrodites overexpressing *pdfr-1* from either a genomic fragment or under the control of a heterologous ASJ-specific promoter. ***p<0.001 as determined by ordinary one-way ANOVA followed by Dunnett's multiple comparisons test. Error bars represent SD. ns, not significant. n = 15 animals for all genotypes except *pdfr-1(ok3425); qdEx151* where n = 10 animals. (**C**) Model for the sex-specific regulation of *daf-7* expression in the ASJ chemosensory neurons by the PDF-1 signaling pathway.

DOI: https://doi.org/10.7554/eLife.36547.007

The following figure supplement is available for figure 4:

**Figure supplement 1.** FISH imaging of endogenous *pdfr-1* mRNA transcripts.

DOI: https://doi.org/10.7554/eLife.36547.008

presented here a set of experiments which implicate the PDF-1 neuropeptide signaling pathway as a critical male-specific regulator of *daf-7* expression in the ASJ neurons. Our data suggest that sexually dimorphic regulation of the PDF-1 receptor, PDFR-1, may serve as a gating mechanism, allowing the ASJ neurons of adult male *C. elegans* to respond to the PDF-1 ligand. We suggest that this ligand-receptor interaction activates a downstream signaling cascade in ASJ terminating in the transcriptional activation of *daf-7*, which in turn promotes male-specific decision-making behaviors (***Figure 4C***, left). We hypothesize that the relative lack of expression or activity of *pdfr-1* in the hermaphrodite ASJ neurons prevents the activation of this pathway and consequently *daf-7* expression is not induced under normal growth conditions in adult hermaphrodites (***Figure 4C***, right). Strikingly, heterologous expression of the PDF-1 receptor in the hermaphrodite ASJ neurons was sufficient to drive *daf-7* expression in an inappropriate physiological context (the hermaphrodite nervous

system). All together, our data suggest that the PDF-1 pathway plays an integral role in facilitating sex-specific differences in gene expression and behavior.

While recent work has revealed sexual dimorphisms at the level of gene expression, neuronal connectivity and neurotransmitter release in the sex-shared nervous system of *C. elegans* (*Hart and Hobert, 2018*; *Hilbert and Kim, 2017*; *Oren-Suissa et al., 2016*; *Pereira et al., 2015*; *Ryan et al., 2014*; *Serrano-Saiz et al., 2017a*, *2017b*; *Weinberg et al., 2018*), the role of neuromodulators and other neuroendocrine signals in facilitating sex-specific responses of neurons in the shared neuronal circuitry has been relatively unexplored. Here, we propose a model in which two pathways, the PDF-1 and DAF-7/TGFβ pathways, act in concert as a neuroendocrine signaling cascade to regulate sex-specific behavior within the context of the sex-shared ASJ neurons (*Figure 4C*). Our data suggest that the PDF-1 pathway functions in tuning the response of the ASJ neurons to this endogenous neuromodulator in a sex-specific manner. Interestingly, recent work in mice has uncovered a similar phenomenon wherein the neuromodulator oxytocin facilitates sex-specific social preference in male mice by modulating the ability of subsets of neurons to respond to social cues (*Yao et al., 2017*). The parallels between this work and ours underscore the role of neuroendocrine signaling through sex-shared nervous system components in shaping sexually dimorphic neuronal activity and behavior in evolutionarily diverse animals.

# Materials and methods

**Key resources table**

| Reagent type (species) or resource | Designation | Source or reference | Identifiers | Additional information |
|---|---|---|---|---|
| Gene (*Caenorhabditis elegans*) | *pdf-1* | NA | WBGene00020317 | |
| Gene (*C. elegans*) | *pdfr-1* | NA | WBGene00015735 | |
| Gene (*C. elegans*) | *daf-7* | NA | WBGene00000903 | |
| Gene (*C. elegans*) | *acy-1* | NA | WBGene00000068 | |
| Genetic reagent (*C. elegans*) | *ksIs2* | PMID: 11677050 | WBTransgene00000788 | *pdaf-7::gfp* |
| Recombinant DNA reagent | Moerman Fosmid Library | Source Bioscience | WRM0629dH07, WRM0627cG01, WRM0641dA07, WRM068aD11 | |
| Recombinant DNA reagent | pPD95.75 | Fire Lab *C. elegans* Vector Kit | Addgene plasmid # 1494 | |
| Recombinant DNA reagent | pCFJ90 | PMID:18953339 | Addgene plasmid # 19327 | *pmyo-2::mCherry*, used as co-injection marker |
| Recombinant DNA reagent | pZH42 | this paper | | *pdf-1* genomic DNA in pUC19 |
| Recombinant DNA reagent | pZH48 | this paper | | *ptrx-1::pdfr-1(cDNA, b isoform, no STOP codon)::F2A::mCherry::unc-54 3'UTR* |
| Recombinant DNA reagent | pZH53 | this paper | | *ptrx-1::ACY-1(P260S)::unc-54 3'UTR* |
| Recombinant DNA reagent | pZH58 | this paper | | *pdfr-1p(distal, 5 kb)::loxP::pdfr-1 cDNA(B isoform)::loxP::unc-54 3'UTR* |
| Recombinant DNA reagent | pZH59 | this paper | | *ptrx-1::nCre* |
| Recombinant DNA reagent | pJDM30 | PMID: 25303524 | | *ptrx-1::daf-7* |
| Recombinant DNA reagent | pSF11 | PMID: 23972393 | | *ptag-168::nCre*, gift of C. Bargmann and S. Flavell |
| Commercial assay or kit | NEBuilder HiFi DNA Assembly Master Mix | NEB | E2621 | |
| Software, algorithm | GraphPad Prism | GraphPad | RRID:SCR_002798 | |
| Other | Alexa Fluor 647 DHS ester | Invitrogen/Thermo Fisher Scientific | A20006 | dye used for conjugation of FISH probes |

### *C. elegans* strains

*C. elegans* strains were cultured as previously described (*Brenner, 1974*; *Hilbert and Kim, 2017*). For a complete list of strains used in this study please see *Supplementary file 1*.

## Cloning and transgenic strain generation

For the *pdf-1* overexpression transgene, a 6.5 kb region of sequence containing the *pdf-1* promoter, coding sequence and 3'UTR were amplified from the fosmid WRM0641dA07 from the Moerman fosmid library. This fragment was cloned into the pUC19 vector backbone by Gibson assembly (*Gibson et al., 2009*) to generate plasmid pZH42. ASJ-specific overexpression of *daf-7* transgenes in the *pdf-1(tm1996)* mutant background were established by reinjection of the plasmid pJDM30 which contains *daf-7* cDNA under the control of the *trx-1* promoter (*Meisel et al., 2014*).

For the ASJ-specific *pdfr-1* rescue construct, the B isoform of the *pdfr-1* cDNA with no stop codon was amplified from cDNA generated with an Ambion RetroScript kit using primers based on previously described annotation of the isoform (*Barrios et al., 2012*). The *trx-1* ASJ-specific promoter was amplified as previously described (*Hilbert and Kim, 2017*). An F2A::mCherry fragment was amplified off a plasmid that was a gift from C. Pender and H.R. Horvitz. All fragments were cloned into the pPD95.75 backbone with an intact *unc-54* 3'UTR by Gibson assembly to generate plasmid pZH48. Genomic rescue of *pdfr-1* was done by injection of the WRM0629dH07 fosmid from the Moerman fosmid library.

Cloning for cell-specific RNAi experiments was carried out similarly to the method previously described (*Esposito et al., 2007*). A 1.2 kb fragment of the ASJ-specific *trx-1* promoter was amplified from the fosmid WRM0627cG01 from the Moerman fosmid library with overlap to either *pdfr-1* or GFP in either the sense or antisense direction. A 1.8 kb exon rich region of the *pdfr-1* coding sequence was amplified from the fosmid WRM0629dH07. A 1 kb fragment containing part of the GFP coding sequence was amplified from the plasmid, pPD95.75. The GFP and *pdfr-1* fragments were cloned in the sense and anti-sense directions with the *trx-1* promoter into the pUC19 plasmid backbone using Gibson Assembly. PCR with nested primers was then used to amplify only the promoter and gene sequence off the plasmid backbone and these PCR products were purified and used for injections. Both sense and anti-sense PCR products were injected at a concentration of 20 ng/μL along with pCFJ90 (*Frøkjaer-Jensen et al., 2008*) at 2.5 ng/μL and 1 kb ladder as carrier DNA.

For the floxed *pdfr-1* rescue strain, the *pdfr-1* cDNA was amplified with primers carrying loxP sequences on either side. The 5 kb *pdfr-1* promoter was amplified from the fosmid WRM068aD11. These fragments were cloned into a pPD95.75 backbone with an intact *unc-54* 3'UTR by Gibson Assembly to generate plasmid pZH58. ASJ-specific Cre lines were generated by swapping the *trx-1* promoter into the plasmid pSF11 (gift of S. Flavell and C. Bargmann) in place of the *tag-168* pan-neuronal promoter to generate plasmid pZH59. Pan-neuronal Cre lines were generated by re-injection of pSF11 at a concentration of 20 ng/μL. For all Cre lines, pCFJ90 was used as a co-injection marker at a concentration of 2.5 ng/μL.

For the ACY-1(gf) construct, the 3.8 kb *acy-1(P260S)* fragment was amplified from genomic DNA extracted from the strain CX15050 (gift from S. Flavell and C. Bargmann) which carries a transgenic array with the *acy-1(P260S)* cDNA under the control of a different promoter. This fragment was cloned into a plasmid backbone carrying the *trx-1* promoter and *unc-54* 3'UTR to generate pZH53. All fosmids and plasmids were verified by sequencing and injected at a concentration of 50 ng/μL along with a plasmid carrying *pofm-1::gfp* at 50 ng/μL as a co-injection marker unless otherwise noted. At least three independent transgenic lines were obtained and analyzed for each construct and one or two representative lines are shown. For a list of all primers used in this paper, please see *Supplementary file 2*.

## Measurement of gene expression in ASI and ASJ neurons

Quantification of *daf-7* expression was performed as described in (*Hilbert and Kim, 2017*) using the *ksIs2(pdaf-7::gfp)* transgene (*Murakami et al., 2001*). All adult quantifications were done on animals 72 hr after egg lay. Quantification of animals on *P. aeruginosa* were performed as before.

### Starvation assays

Starvation assays and measurement of *pdaf-7::gfp* fluorescence in the ASJ neurons of starved males was performed as previously described (*Hilbert and Kim, 2017*).

### Mate-Searching assays

Mate-searching assays were performed as previously described (*Hilbert and Kim, 2017*; *Lipton et al., 2004*).

### *P. aeruginosa* lawn avoidance assays

*P. aeruginosa* plates were prepared as described in (*Hilbert and Kim, 2017*). Animals were synchronized by treatment with bleach and allowed to hatch and arrest as L1 larvae before being dropped onto *E. coli* plates. L4 animals were transferred to the center of the *P. aeruginosa* lawn, incubated at 25°C and then scored for avoidance after 16 hr.

For male lawn avoidance assays shown in *Figure 1—figure supplement 2B*, plates and animals were prepared using the same method as for hermaphrodites, but males were placed individually onto plates seeded with *P. aeruginosa* as L4s. Plates were incubated at 25°C and then scored for avoidance after 16 hr. These experiments were repeated three times, with 30 individual animals per genotype in each replicate.

### Fluorescence In Situ Hybridization

FISH was performed as previously described (*Hilbert and Kim, 2017*; *Raj et al., 2008*). The *pdfr-1* probe was constructed by pooling together 36 unique 20 nucleotide oligos that tile across basepairs 580–1540 in the *pdfr-1* B-isoform cDNA. This sequence is contained in all isoforms of *pdfr-1* so should anneal to any endogenous *pdfr-1* mRNA. After pooling, oligos were coupled to Alexa Fluor 647 NHS ester (Invitrogen/Thermo Fisher Scientific) and then purified by HPLC.

### Statistical analysis

All statistical analysis was performed using the Graphpad Prism software. Statistical tests used for each experiment are listed in the figure legend.

## Acknowledgements

We thank C Bargmann, S Flavell, HR Horvitz, J Kaplan, and the *Caenorhabditis* Genetics Center (which is funded by the NIH Office of Research Infrastructure Programs P40 OD010440) for strains and reagents. We thank S Flavell and members of the Kim and Horvitz labs for helpful discussions in the development of the manuscript.

## Additional information

### Funding

| Funder | Grant reference number | Author |
| --- | --- | --- |
| National Institutes of Health | GM084477 | Dennis H Kim |
| National Institutes of Health | T32GM007287 | Zoë A Hilbert |

The funders had no role in study design, data collection and interpretation, or the decision to submit the work for publication.

### Author contributions

Zoë A Hilbert, Conceptualization, Formal analysis, Investigation, Visualization, Methodology, Writing—original draft, Writing—review and editing; Dennis H Kim, Conceptualization, Supervision, Funding acquisition, Visualization, Writing—review and editing

**Author ORCIDs**
Zoë A Hilbert (iD) http://orcid.org/0000-0002-3833-6912
Dennis H Kim (iD) https://orcid.org/0000-0002-4109-5152

**Decision letter and Author response**
Decision letter https://doi.org/10.7554/eLife.36547.013
Author response https://doi.org/10.7554/eLife.36547.014

## Additional files

**Supplementary files**
• Supplementary file 1. *C. elegans* strains used in this study. A comprehensive list of the strains used in this study. Strain source (this study or others) is indicated.
DOI: https://doi.org/10.7554/eLife.36547.009

• Supplementary file 2. Oligos used for transgenic strain construction and *pdfr-1* FISH A comprehensive list of the oligos created for this study. All sequences are listed in the 5' to 3' direction.
DOI: https://doi.org/10.7554/eLife.36547.010

• Transparent reporting form
DOI: https://doi.org/10.7554/eLife.36547.011

## Data availability

All data generated or analysed during this study are included in the manuscript and supporting files.

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
