## [Decision Letter]

Thank you for submitting your article "A sexually dimorphic neuroendocrine cascade acts through shared sensory neurons to regulate behavior in *C. elegans*" for consideration by *eLife*. Your article has been reviewed by two peer reviewers, and the evaluation has been overseen by Oliver Hobert as the Reviewing Editor and a Senior Editor. The following individual involved in review of your submission has agreed to reveal her identity: Piali Sengupta (Reviewer #1).

The reviewers have discussed the reviews with one another and the Reviewing Editor has drafted this decision to help you prepare a revised submission.

In brief, both reviewers appreciate the value of this follow-up study, but they also agree that a set of additional experiments is required to firm up the conclusions of this study. You can see their comments in detail below. In summary, the following set of experiments are essential:

1) Proper quantification of smFISH experiments. Examination at different stages.

2) To show necessity, remove *pdfr-1* specifically from ASJ and assess effects on *daf-7* expression.

3) Clarify the placement of *acy-1* in the pathways regulating *daf-7* expression: are the effects of *acy-1(gf)* suppressed by starvation? Does *acy-1(lf)* block the effect of PA14 or *pdfr-1* overexpression in hermaphrodites?

4) Test whether *daf-7* expression in ASJ suppresses the leaving phenotype of *pdfr-1* or *pdf-1* mutants.

Less important: show the behavioural phenotype of *pdf-1* and *pdfr-1* mutant males in the pathogenic bacteria avoidance assay like shown for hermaphrodites (Figure 1E).

*Reviewer #1:*

This is nice follow up to the authors' previous paper and is appropriate for publication as a Research Advance with a few relatively small revisions.

1) The FISH data shown in Figure 3C need to be quantified in both males and hermaphrodites.

2) The effects of *acy-1(gf)* on *daf-7* expression in both males and hermaphrodites is intriguing. While the interpretation that *acy-1* acts downstream of *pdfr-1* is consistent with these data, it's also possible that multiple pathways – including those induced by *Pseudomonas* or blocked by starvation – might act via cAMP production. For example, cAMP (Khanna Sci Reports, 2016) can affect *daf-7* expression in ASI. Might the manipulation of cAMP levels via *acy(gf)* be engaging a common regulatory mechanism for *daf-7* expression? Given the hierarchical nature of the starvation signal, one could expect that the effects of *acy-1(gf)* would be blocked in starved males if this is indeed specific to the PFDR-1 pathway. Alternatively, *acy-1* mutations may block the effect of PA14 or *pdfr-1* overexpression in hermaphrodites, especially considering *acy-1* mutants have been previously reported to have a pathogen susceptibility phenotype (Alper, 2008). It would be useful to perform a few simple experiments here to clarify the placement of *acy-1(gf)* in the hierarchy and to strengthen the conclusions of this work.

3) Along similar lines, does *acy-1(gf)* or *daf-7* expression in ASJ suppress the leaving phenotype of *pdfr-1* or *pdf-1* mutants?

4) The authors may wish to modify the model figure to clearly indicate that the inclusion of *acy-1* is based on inferring the effects of *acy(gf)* overexpression, especially based on point #2 above.

*Reviewer #2:*

In this manuscript, Hilbert and Kim report that male-specific expression of *daf-7* in ASJ is regulated by the neuropeptide pathway PDF acting directly on ASJ. These findings expand on their previous work showing that *daf-7* expression in ASJ is regulated in a sex-specific and age-specific manner to regulate mate-searching behaviour in males.

The authors present convincing evidence that PDF signalling regulates the expression of *daf-7* specifically in ASJ and specifically in males. However, what is not so convincing to me is that *daf-7* expression is regulated cell-autonomously, through male-specific expression of *pdfr-1* in ASJ.

The model that they propose is not fully consistent with some of their previous findings. In their previous paper they carried out sex-transformation experiments which showed that *daf-7* expression in ASJ requires ASJ to be masculine but that masculinization of ASJ alone is not sufficient for *daf-7* to come on. These results indicate that there is cell non-autonomous regulation of *daf-7* expression. In the current manuscript, *daf-7* expression can be induced in hermaphrodites by forcing expression of *pdfr-1* in ASJ, indicating that the source of PDF is not sexually dimorphic. These two sets of data are more consistent with the model that PDF regulates *daf-7* expression in ASJ cell non-autonomously, by acting on other *pdfr-1*-expressing neurons that are sexually dimorphic.

The evidence that the authors present for cell-autonomous effect of *pdfr-1* on ASJ is that *daf-7* expression in ASJ can be induced in both males and hermaphrodites by overexpressing *pdfr-1* in ASJ, but this could be an overexpression phenotype that does not reflect the endogenous mechanism. Stronger evidence in support of a direct action of PDF on ASJ would be to show necessity of *pdfr-1* specifically in ASJ. This should be done by removing *pdfr-1* only in ASJ (by cell specific RNAi for example).

The second piece of evidence they provide is the detection of *pdfr-1* mRNA in the ASJ of males by smFISH, but the smFISH experiments are not totally convincing. The smFISH experiments should be quantified: what proportion of males express *pdfr-1* in ASJ? And what proportion of hermaphrodites? They only show 3 photos, and in one photo of hermaphrodite ASJ, I can see what looks to me like puncta. The overlap of smFISH with the ASJ labelled with GFP for identification should also be shown.

Additional comments:

In their previous paper, they showed that *daf-7* expression in the ASJ of males is regulated by age (only ON in adults). A better understanding of the temporal regulation of the PDF pathway should be provided. Is *pdfr-1* expressed in ASJ only in adult males or earlier as well? At what stage does *daf-7* come on in the ASJ of hermaphrodites and males that ectopically overexpress *pdfr-1* in ASJ from early larval stages?

Another conclusion of the manuscript is that PDF regulates mate searching through the expression of *daf-7* in ASJ. This is based on the observation that the *daf-7* mutation suppresses the enhanced food leaving induced by PDF overexpression. But, can the food leaving defects of *pdfr-1* mutants be rescued by expression of *pdfr-1* or *daf-7* only in ASJ?

The authors should show the behavioural phenotype of *pdf-1* and *pdfr-1* mutant males in the pathogenic bacteria avoidance assay like they do for hermaphrodites (Figure 1E).

---

## [Author Response]

In brief, both reviewers appreciate the value of this follow-up study, but they also agree that a set of additional experiments is required to firm up the conclusions of this study. You can see their comments in detail below. In summary, the following set of experiments are essential:1) Proper quantification of smFISH experiments. Examination at different stages.2) To show necessity, remove pdfr-1 specifically from ASJ and assess effects on daf-7 expression.

We thank the reviewers for the suggestion to further assess the necessity of *pdfr-1* function in ASJ for the regulation of *daf-7* expression. We performed several experiments to address these concerns as outlined below:

As was suggested, we generated transgenic strains carrying ASJ-specific RNAi against *pdfr-1* and also against GFP as a positive control. Cell-specific RNAi of either GFPor *pdfr-1* leads to attenuated *pdaf-7::gfp* fluorescence in the ASJ neurons of adult males. We show this data for two independent lines for each condition (Figure 2C of revised manuscript) but observed similar decreased expression in a number of additional transgenic lines for both *pdfr-1* and *gfp* knockdown. We feel that this data clearly suggests a role for *pdfr-1* function in ASJ to cell-autonomously regulate *daf-7* expression, and we have revised the manuscript to include discussion of these results as well as included this data as Figure 2C.

Additionally, we generated transgenic strains with floxed *pdfr-1* cDNA under the control of a 5 kb promoter region reported by Flavell et al. (2013) to rescue roaming defects in *pdfr-1* mutant hermaphrodites. These strains have partial rescue of *daf-7* expression in ASJ, and we observed that this rescue is abolished by Cre recombinase expression either pan-neuronally or in ASJ alone. The ability of ASJ-specific Cre to abolish the rescuing effects of this floxed *pdfr-1* transgene further supports the idea that *pdfr-1* functions specifically in ASJ to regulate *daf-7* expression in males. We have included this data as Figure 2D in our revised manuscript and included discussion of these results in the revised text.

Upon further examination of *pdfr-1* mRNA by FISH, we note that fluorescent puncta are clearly visible in 20-30% of adult male animals across multiple experiments, but we have not observed the same in hermaphrodites. Given that there are two distinct populations of male animals, quantifications of probe fluorescence across both populations combined show no significant difference between males and hermaphrodites (see newly added Figure 4—figure supplement 1F). We also do not see evidence for *pdfr-1* expression in L4 males. We hypothesize that *pdfr-1* expression in ASJ may be low, or temporally or spatially restricted making it difficult for us to capture a moment in which the majority of male animals have clear evidence of expression. Nevertheless, a more definitive analysis of *pdfr-1* expression would be required to establish sexually dimorphic expression of this gene. As such, we have significantly revised the manuscript and our conclusions to reflect this and have moved all of the FISH data from main text Figure 3 to Figure 4—figure supplement 1. We have also provided our quantification in the revised manuscript as Figure 4—figure supplement 1F. Although detection of *pdfr-1* expression is challenging, our aforementioned strong functional data suggesting that PDFR-1 activity in ASJ is necessary and sufficient to drive *daf-7* expression supports our model that *pdfr-1* acts sexually dimorphically in ASJ to regulate *daf-7* expression in males.

3) Clarify the placement of acy-1 in the pathways regulating daf-7 expression: are the effects of acy-1(gf) suppressed by starvation? Does acy-1(lf) block the effect of PA14 or pdfr-1 overexpression in hermaphrodites?

We analyzed the effect of starvation on *daf-7* expression in male animals carrying the ASJ specific *acy-1(gf)* array and we observed that starvation suppresses *daf-7* expression in the ASJ neurons of these animals (Figure 1—figure supplement 1A). We interpret this to mean that *acy-1* and cAMP production function independently from starvation in the regulation of *daf-7* expression in the ASJ neurons. This data has been included in our revised manuscript as Figure 1—figure supplement 1 and we have added discussion of our interpretation of this in the revised text.

4) Test whether daf-7 expression in ASJ suppresses the leaving phenotype of pdfr-1 or pdf-1 mutants.

As the reviewers suggested, we looked to see if expression of *daf-7* could rescue the leaving defects of the *pdf-1* mutant males. Additionally, we looked to see if mutation of the antagonistic co-SMAD in the *daf-7*/TGF-β pathway, DAF-3, could suppress these leaving defects as it does for the *daf-7* mutants (see our previous *eLife* paper, Hilbert and Kim, 2017). In all of these strains, we saw no suppression of the leaving defects of the *pdf-1* mutant males.

Given our results with the PDF-1 overexpression lines and the dependence of their leaving behavior on *daf-7* function shown in Figure 1 of our manuscript, we hypothesize that the PDF-1 pathway functions through *daf-7* but also through parallel mechanisms to regulate leaving behavior in male animals and that *daf-7* expression in ASJ is necessary but not sufficient for leaving behaviors. We have included some of this data as Figure 1—figure supplement 1B and also provided a clarification in the text of the interaction between these two pathways with regards to mate-searching behaviors in males.

Less important: show the behavioural phenotype of pdf-1 and pdfr-1 mutant males in the pathogenic bacteria avoidance assay like shown for hermaphrodites (Figure 1E).

We thank the reviewers for this suggestion. Given the propensity of male animals to aggregate, we performed a modified pathogen avoidance assay with a single male per PA14 plate to assess the avoidance phenotypes of the *pdf-1* and *pdfr-1* mutant males. As with hermaphrodites, we saw no difference in the ability of the mutant males to avoid the pathogenic lawn compared to wild type animals. These data are now included as Figure 1—figure supplement 2B and details of the methods used for this modified assay have been included in our revised manuscript.